# Spatial regulation of thermomorphogenesis by HY5 and PIF4 in Arabidopsis

Sanghwa Lee [1], Wenli Wang [1] & Enamul Huq [1✉]

Plants respond to high ambient temperature by implementing a suite of morphological changes collectively termed thermomorphogenesis. Here we show that the above and below ground tissue-response to high ambient temperature are mediated by distinct transcription factors. While the central hub transcription factor, PHYTOCHROME INTERCTING FACTOR 4 (PIF4) regulates the above ground tissue response, the below ground root elongation is primarily regulated by ELONGATED HYPOCOTYL 5 (HY5). Plants respond to high temperature by largely expressing distinct sets of genes in a tissue-specific manner. HY5 promotes root thermomorphogenesis via directly controlling the expression of many genes including the auxin and BR pathway genes. Strikingly, the above and below ground thermomorphogenesis is impaired in *spaQ*. Because SPA1 directly phosphorylates PIF4 and HY5, SPAs might control the stability of PIF4 and HY5 to regulate thermomorphogenesis in both tissues. These data collectively suggest that plants employ distinct combination of SPA-PIF4-HY5 module to regulate tissue-specific thermomorphogenesis.

[1] Department of Molecular Biosciences and The Institute for Cellular and Molecular Biology, The University of Texas at Austin, Austin, TX, USA.
✉email: huq@austin.utexas.edu

High ambient temperature due to global warming affects all terrestrial life forms including crop yield[1]. Being rooted in soil, sessile plants respond to high ambient temperature by changing their architecture in a developmental process termed thermomorphogenesis, which is characterized by elongated hypocotyl, petiole and root tissues, hyponastic growth, reduced stomatal density and early flowering[2–4]. These morphological changes allow plants to cope with and complete reproductive cycle under high ambient temperature.

Plants perceive high ambient temperature using diverse mechanisms. These include the red-light photoreceptor phytochrome B (phyB)[5,6], an RNA thermoswitch within the 5′-untranslated region of the *PHYTOCHROME INTERCTING FACTOR 7* (*PIF7*)[7] and a prion-like domain in EARLY FLOWERING 3 (ELF3)[8]. Downstream of the thermosensors, PIF4 acts as a central hub transcription factor controlling target genes especially auxin biosynthesis and signaling genes to promote hypocotyl and petiole elongation[9–12]. In addition, a number of factors including Cryptochrome 1 (Cry1), ELF3, TOC1, HMR, BBX18/23, TCP, FCA, SPAs, HECATE1/2 (HEC1/2), COP1-DET1-HY5 module, and GI-RGA-PIF4 module have been shown to regulate thermomorphogenesis[13–25]. All of these factors impinge upon PIF4 either controlling the PIF4 abundance and/or activity to regulate thermomorphogenesis.

High ambient temperature also affects root system architecture[26,27], which is linked to crop yield[28,29]. In sunflower and cotton, elevated temperature increases lateral root length and number[27]. In Arabidopsis, elevated temperature has been shown to increase root elongation by regulating Brassinosteroid (BR) signaling[30]. In addition, several reports showed that auxin signaling is also involved in root elongation under normal and high ambient temperature conditions[31–33]. For example, Feraru et al. showed that PIN-LIKES 6 (PILS6) acts as a negative regulator of root growth, and high ambient temperature-mediated destabilization of PILS6 abundance increases the nuclear availability of auxin to promote root elongation[32]. In addition, a recent report also showed that the shoot and root thermomorphogenesis are coordinated, and a shoot derived HY5/PIF regulatory module controls auxin biosynthetic and signaling genes to regulate root thermomorphogenesis[34].

Our understanding of thermomorphogenesis is mostly based on the above ground tissues (e.g., hypocotyl and petiole elongation, flowering time)[3]. However, the mechanistic details of how the below-ground tissues of plants respond to high ambient temperature is still rudimentary. Here we show that ELONGATED HYPOCOTYL 5 (HY5) acts as a key transcription factor in root thremomorphogenesis. The expression and stability of PIF4 and HY5 are spatially regulated by high ambient temperature. In addition, SPA kinases are necessary to stabilize both PIF4 and HY5 to regulate thremomorphogenesis in a tissue-specific manner.

## Results

**Analyses of spatial thermomorphogenesis at seedlings stage**. To examine spatial regulation of thermomorphogenesis, we grew wild type (WT) Arabidopsis seedlings at either 22 °C or 28 °C under constant light (CL) up to seven days and measured the root and hypocotyl lengths over time (Fig. 1a, b). Thermo-induced primary root elongation occurred gradually over time (Fig. 1b, c) and the primary root length ratio of 28/22 remained similar (Fig. 1c, d). Primary root length was also elongated at 28 °C under long day (LD) conditions similar to that in CL, while the primary root length was even shorter at 28 °C compared to 22 °C under short day (SD) conditions, consistent with the previous report[30] (Supplementary Fig. 1). In contrast, thermo-induced hypocotyl

elongation occurred over time, and the hypocotyl length ratio of 28/22 increased over time and saturated at around five days (Fig. 1e, f). To further examine whether the thermo-induced primary root elongation can be observed under natural soil-grown conditions, we grew wild-type Arabidopsis seedlings for three days on MS plate at 22 °C and then transferred to soil for additional five days at either 22 °C or 28 °C under LD conditions and measured the root length (Supplementary Fig. 2a–c). Similar to the growth conditions on MS plate, soil-grown seedlings displayed elongated roots at high ambient temperature. The hypocotyl thickness was also measured using light microscopy under the continuous light conditions (Supplementary Fig. 3a). The data show that the hypocotyl diameter was increased ~30% after 4 days of treatment at 28 °C (Supplementary Fig. 3b), indicating that high ambient temperature promotes the thickness as well as elongation of hypocotyl. Overall, high ambient temperature affects seedling architecture of both above ground and below-ground tissues profoundly.

**HY5 is necessary for root thermomorphogenesis**. PIF4 has been described as a key component in the hypocotyl and petiole elongation under high ambient temperature[9]. However, the thermo-induced root elongation was not affected in *pif4* background[30]. To identify the factor(s) necessary for root thermomorphogenesis, we analyzed the primary root length phenotype of several mutants and overexpression lines under high ambient temperature (Fig. 2a–c). As expected, *phyB-9* did not respond to high ambient temperature, whereas *PHYB* over-expression line showed similar response to WT, suggesting that phyB regulates both above and below-ground tissues at high ambient temperature. We also analyzed the primary root length phenotype of *pif4-2* and *pifQ* at high ambient temperature to examine if PIFs are involved in root thermomorphogenesis (Fig. 2a, b, Supplementary Fig. 4). The data show that both *pif4-2* and *pifQ* responded to root thermomorphogenesis with similar primary root length ratio of 28/22, indicating that PIFs do not contribute to primary root elongation at high ambient temperature Supplementary Fig. 4a–c. Because HY5 has been shown to regulate thermomorphogenesis as well as root hair elongation[14,35], we examined root thermomorphogenesis phenotype of *hy5-215* at high ambient temperature. The results show that the primary root length of *hy5-215* mutant did not alter at high ambient temperature, indicating that HY5 is necessary for the primary root elongation under high ambient temperature. Similar to the MS-grown seedlings, *hy5-215* mutant did not show root thermomorphogenesis, while *pifQ* mutant responded to high ambient temperature in soil-grown conditions (Supplementary Fig. 2). Furthermore, to elucidate the genetic relationship of *PIFs* and *HY5* in root thermomorphogenesis, the primary root length was measured for *hy5-215 pifQ* at high ambient temperature. As expected, the primary root length of *hy5-215 pifQ* did not respond to high ambient temperature, whereas *pifQ* displayed similar root length to that of wild-type (Fig. 2a–c), suggesting that *hy5-215* is epistatic to *pifQ* for root thermomorphogenesis.

When we examined the hypocotyl thickness phenotype under high ambient temperature. Both *hy5-215* and *pifQ* did not show any difference, while the WT displayed an increase in the hypocotyl thickness in response to high ambient temperature (Supplementary Fig. 5a–c). *PIF4* overexpression line displayed an increased thickness constitutively compared to WT, while *HY5* overexpression in the *hy5* background complemented the phenotype. Moreover, epistatic analyses showed that the hypocotyl thickness phenotype of *hy5-215 pifQ* is similar to that of *pifQ* and *hy5-215* (Supplementary Fig. 5a–c), suggesting that both PIFs and HY5 are acting in the same genetic pathway to

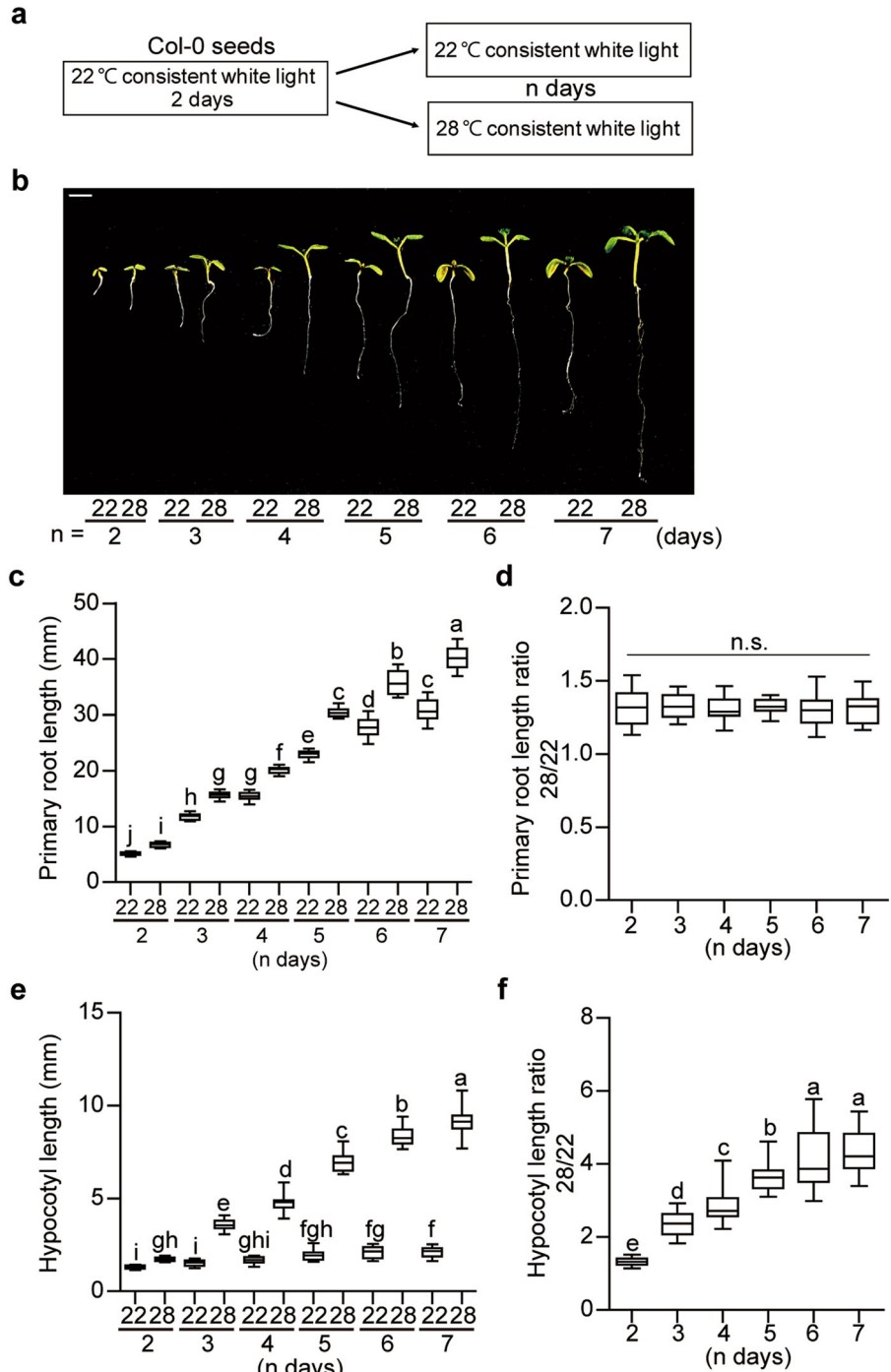

**Fig. 1 Altered seedling architecture at high ambient temperature. a** Simplified experimental condition used in this study for phenotypic analysis. **b** Photograph showing seedling phenotypes of Columbia-0 grown at 22 °C and 28 °C over time. Seedlings were grown for two days in continuous white light at 22 °C and then either kept at 22 °C or transferred to 28 °C for additional n days before being photographed. The scale bar represents 2 mm. **c, e** Bar graphs show the primary root (**c**) or hypocotyl (**e**) lengths for seedlings grown under conditions described in (**a**). **d, f** For the length ratio 28/22, primary root or hypocotyl length at 28 °C is divided by those values at 22 °C. The letters a–j indicate statistically significant differences between means of primary root or hypocotyl lengths ($n = 15$, $P < 0.05$) based on one-way ANOVA analyses with Tukey's HSD test. n.s. stands for not significant. For the box and whisker plots in (**c**-**f**), the boxes represent from the 25th to the 75th percentile and the bars equal the median values. The source data underlying the root (**c, d**) and hypocotyl (**e, f**) measurements are provided in the Source data file.

regulate the hypocotyl thickness in response to high ambient temperature. Overall, these data suggest that HY5 is a key transcription factor for root thermomorphogenesis, while both PIFs and HY5 are acting in the hypocotyl tissue to regulate elongation as well as thickness in response to high ambient temperature.

**HY5 controls global gene expression at elevated temperature.** In thermomorphogenesis, an alteration in global gene expression has been reported mostly using whole seedlings[6,19,24]. To examine whether global gene expression is altered in a tissue-specific manner at high ambient temperature, WT and *hy5-215* seedlings were grown at 22 °C for 6 days under constant white light, and

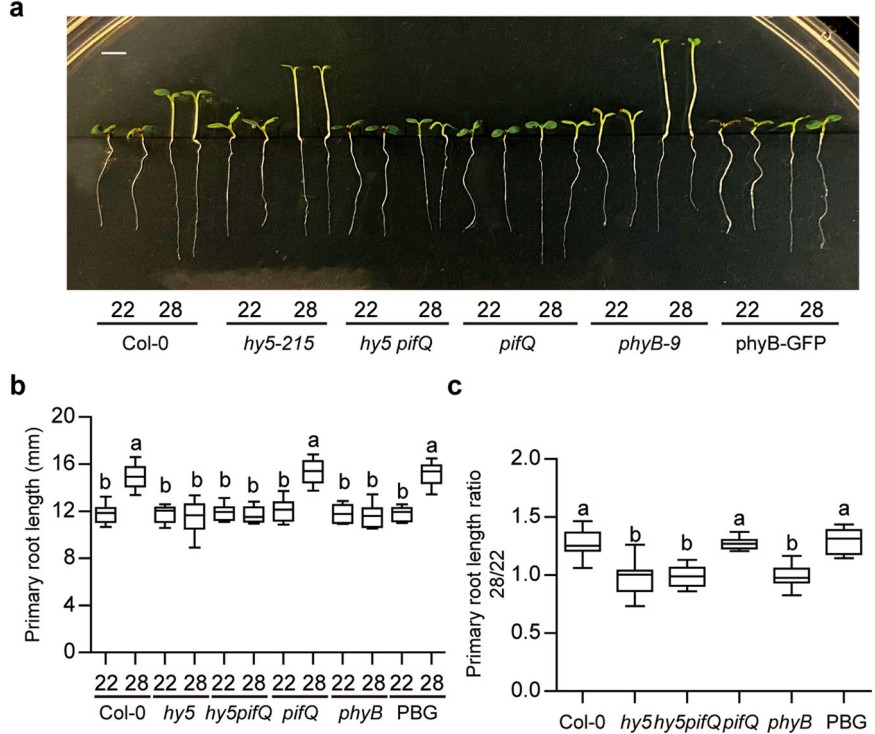

**Fig. 2 HY5 is essential for root thermomorphogenesis. a** Photograph showing seedling phenotypes of different mutants grown at 22 °C and 28 °C. Seedlings were grown for 2 days in continuous white light at 22 °C and then either kept at 22 °C or transferred to 28 °C for additional 4 days before being photographed. The scale bar represents 2 mm. PBG stands for *35S:phyB-GFP*. **b, c** Bar graphs show the primary root lengths for seedlings grown under conditions described in (**a**). For the length ratio 28/22 (**c**), primary root length at 28 °C is divided by that at 22 °C. The letters a, b indicate statistically significant differences between means of primary root lengths ($n = 15$, $P < 0.05$) based on one-way ANOVA analyses with Tukey's HSD test. For the box and whisker plots in (**b, c**), the boxes represent from the 25th to the 75th percentile and the bars equal the median values. The source data underlying the root measurements (**b, c**) are provided in the Source data file.

then exposed to 28 °C for additional 4 h before the shoot (hypocotyl and cotyledon) and root samples were collected. RNAseq analysis was performed using total RNA isolated from these samples. The results show that the shoot and root samples respond to high ambient temperature by differentially expressing genes (DEGs) that are largely distinct with partial overlap (Supplementary Fig. 6a, Supplementary data 1). Gene Ontology (GO) analysis of the shared 228 DEGs between the shoot and root showed an enrichment of temperature/heat/stress response genes including *HSP90* (Supplementary Fig. 6b)[33], suggesting that both organs respond to elevated temperature at least partially by controlling a common set of genes. However, the genes responding to high temperature in an organ-specific manner show an enrichment of different classes. For example, while genes involved in regulation of protein kinase activity, response to nitrogen compounds, mitotic cell cycle, response to abiotic stimulus show an enrichment in the shoot tissue (Supplementary Fig. 6c), the genes involved in CDP-diacylglycerol metabolic process, secondary metabolite biosynthetic process, cellular response to oxygen levels and oxidative stress are enriched in the root tissue (Supplementary Fig. 6d). These data suggest that the above ground and below-ground tissues respond to high ambient temperature differently.

When comparing the shoot samples from the WT Col-0 and *hy5-215* in response to high ambient temperature, 1427 and 1083 DEGs were identified in the Col-0 and *hy5-215*, respectively, with 418 genes shared between the Col-0 and *hy5-215* shoot samples (Fig. 3a). Importantly, >70% of WT DEGs are HY5-dependent, supporting the previous reports that HY5 contributes to shoot thermomorphogenesis[13,36]. By contrast, 1129 and 846 genes were

differentially expressed in the root samples of Col-0 and *hy5-215*, respectively, in response to high ambient temperature. In the root tissue, >84% of Col-0 DEGs are HY5-dependent, suggesting a prominent role of HY5 in root thermomorphogenesis. Figure 3b displays the heatmap analysis of the 1427 and 1129 DEGs from the WT Col-0 hypocotyl and root samples, respectively, in both WT and *hy5-215*, indicating that the gene expression is globally regulated by HY5 in both hypocotyl and root tissues. GO analysis of the HY5-dependent 1009 DEGs in the hypocotyl, and 956 DEGs in the root samples showed an enrichment of diverse processes that are largely distinct between the shoot and root tissues (Fig. 3c), suggesting that HY5 might regulate different pathways in a tissue-specific manner to respond to high ambient temperature.

We also compared our HY5-regulated genes to those of the HY5-bound genes that has been recently described[37]. These analyses show that >50% of the HY5-regulated genes are also HY5-direct target genes (Supplementary Fig. 7a). When we compared at the organ-specific manner, >51% of the HY5-regulated genes in the shoot tissue are also HY5 direct target genes, while in the root sample, >50% of the HY5-regulated genes are HY5 direct target genes (Supplementary Fig. 7a). Only 55 of the 939 potentially HY5-regulated and HY5-direct target genes are expressed in both organs (Supplementary Fig. 7a), suggesting that HY5 regulates a large proportion of genes in an organ-specific manner to respond to high ambient temperature.

To confirm our RNAseq data, we selected 3 genes (*IAA19*, *SAUR40*, and *SAUR77*) involved in auxin signaling for independent verification by qPCR, as these genes have been linked to both root and hypocotyl elongation[38–41]. Consistent with our RNAseq

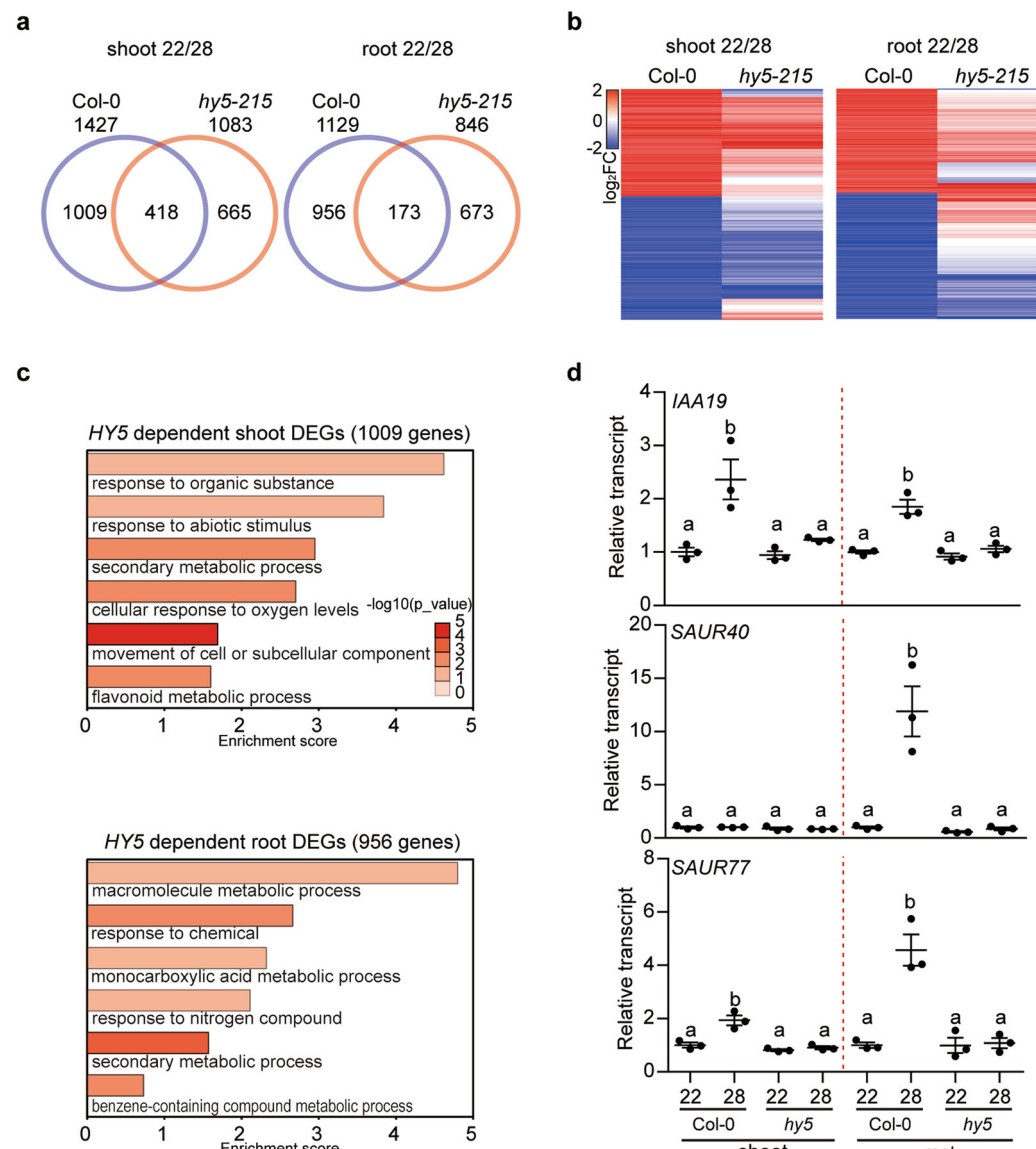

**Fig. 3 HY5 controls global gene expression at elevated temperature in both shoot and root. a** Venn diagrams show tissue-specific (shoot and root) Differentially Expressed Genes (DEGs) in wild-type (WT) vs *hy5-215* mutant at high ambient temperature. Six-day-old white light-grown seedlings were transferred to 22 °C or 28 °C for additional 4 h and total RNA was extracted from three biological replicates for RNA-seq analyses. **b** Heatmap analysis shows 1427 and 1129 DEGs between wild-type and *hy5-215* mutant shoot and root, respectively. **c** Gene Ontology (GO) analysis of HY5-dependent shoot and root DEGs. DEGs were identified with ≥2-fold and FDR ≤ 0.05. **d** RT-qPCR analysis using *IAA19*, *SAUR40*, and *SAUR77*. RT-qPCR samples were from Col-0 whole seedlings, shoot or root grown for 6 days at 22 °C and then either kept at 22 °C or transferred to 28 °C for 4 h. Three biological repeats were performed. Error bars indicate SD (*n* = 3). Relative gene expression levels were normalized using the expression level of *ACT7* and the values of those genes in wild-type. The letters a, b indicate statistically significant differences (*n* = 3, *P* < 0.05) based on one-way ANOVA analyses with Tukey's HSD test.

data (Supplementary Fig. 7b), the transcript levels of these genes were highly repressed in *hy5-215* in both shoot and root samples (Fig. 3d). Strikingly, *SAUR40* was highly expressed only in the WT root sample, indicating a root-specific HY5 regulation of *SAUR40* at high ambient temperature (Fig. 3d, middle). These data are consistent with the previous report showing the SAUR40-EYFP expression in the root tip[38]. Since the auxin signaling genes (*IAA19*, *SAUR40*, *SAUR77*, and *PILS6*) are potentially direct targets of HY5 as the ChIPseq assays showed an enrichment of HY5 binding to the promoter regions of these genes (Supplementary Fig. 7c)[37], we performed Chromatin Immunoprecipitation (ChIP) assays using 35S:HY5-GFP/*hy5-215* whole seedlings (Supplementary Fig. 7d). The result shows that the HY5 binding to these promoters was enriched at high

ambient temperature, indicating that HY5 directly binds to the promoter regions of *IAA19*, *SAUR40*, and *SAUR77*. Taken together, these data suggest that HY5 promotes root thermomorphogenesis by regulating gene expression globally in root tissue.

**HY5 promotes BR-mediated root elongation at high temperature.** A recent study showed that the primary root elongation at high ambient temperature is also mediated by the brassinosteroid (BR) signaling pathway[30]. Therefore, we examined whether the thermo-insensitive primary root phenotype of *hy5-215* mutant is caused in part by an alteration in the expression of BR metabolic genes. Although these genes were not identified in our RNAseq analyses, qRT-PCR assays show that the transcript

level of the BR biosynthetic gene *CPD* was upregulated, while that of BR catabolic genes *BAS1* and *SOB7* was downregulated in the WT root tissue at high ambient temperature as previously reported[30]. However, the transcript level of these genes did not alter in the *hy5-215* mutant in response to high ambient temperature (Supplementary Fig. 8a). In contrast, the expression of these genes was restored in the 35S:HY5-GFP/*hy5-215* complemented line, supporting that HY5 is essential for triggering thermo-induced regulation of the BR metabolic pathway (Supplementary Fig. 8a). Furthermore, because HY5 is a bZIP protein transcription factor that binds to several DNA motifs such as G-box (CACGTG), CG hybrid (GACGTG), CA hybrid (GACGTA) or ACGTG motifs[37,42,43], we searched the shared sequence of these motifs (ACGT motif) within the *CPD*, *BAS1*, and *SOB7* promoter regions. Among the three genes, the promoter regions of *BAS1* and *SOB7* contain multiple HY5-binding sites within the 2 kb promoter regions (3 in *BAS1* and 8 in *SOB7*) (Supplementary Fig. 8b). Therefore, we performed ChIP assays using 35S:HY5-GFP overexpressed in the *hy5-215* mutant background to examine whether HY5 directly binds to the promotor region of these genes. As expected, an enrichment of the HY5 binding was observed in the promoter region containing the ACGT sites (P2 in *BAS1* and P3 in *SOB7*) in response to high ambient temperature compared to the control regions (P1 in *BAS1* and *ACT7*), indicating that HY5 directly binds to the BR metabolic genes (*BAS1* and *SOB7*) and controls their expression. Overall, these data suggest that HY5 might also regulate root thermomorphogenesis by controlling the BR signaling pathway.

**Phosphorylation of HY5 promotes root thermomorphogenesis.** To examine whether HY5 protein level is regulated by high ambient temperature, we used the *HY5-GFP* overexpression line driven by the constitutive cauliflower mosaic virus (CaMV) 35S promoter in the *hy5-215* mutant background. While the *HY5-GFP* transcript level is not altered at high ambient temperature (Supplementary Fig. 9), immunoblot analysis showed that HY5-GFP accumulated at high ambient temperature in both shoot and root tissues of WT seedlings (Supplementary Fig. 10). These data suggest that the altered HY5 protein level might be due to post-translational regulation.

Recently, we have shown that SPA1 can directly phosphorylate HY5 at serine 36 residue and regulate its abundance and activity, consistent with the previous reports[44,45]. These reports showed that the unphosphorylated HY5 is more active but also more unstable, because it is a preferred substrate for COP1-mediated degradation. Since HY5 abundance is post-translationally regulated, we hypothesized that the phosphorylation of HY5 might regulate HY5 stability and activity. Therefore, we analyzed two additional forms of HY5 fused with GFP; phospho-null HY5-S36A, and phospho-mimic HY5-S36D overexpression lines driven by the CaMV35S promoter in the *hy5-215* mutant background[44]. Analyses of the root thermomorphogenesis phenotype showed that all three transgenic plants display similar primary root length at 22 °C. However, the phospho-null HY5-S36A form did not respond to high ambient temperature, whereas the phospho-mimic HY5-S36D form showed longer primary root length at high ambient temperature similar to the wild-type HY5 overexpression line (Fig. 4a–c). These data suggest that phosphorylation of HY5 is essential for root thermomorphogenesis. To check whether HY5 protein level is altered at high ambient temperature in a tissue-specific manner, western blot was performed (Supplementary Fig. 10). While three different HY5 forms show similar level at normal condition, WT HY5-GFP and HY5-S36D were more abundant, whereas HY5-S36A showed decreased level at high ambient temperature in both shoot and

root tissues. These data show that phosphorylation of HY5 is essential for thermo-induced stabilization. To further analyze whether HY5 protein level is regulated in a tissue-specific manner, fluorescent microscopy was used to detect GFP signal in the root tissues (Fig. 4d, e). Similar to the immunoblot (Supplementary Fig. 10), wild-type HY5-GFP signal was accumulated in the entire root tissue at high ambient temperature (Fig. 4d, e). However, HY5-S36A showed reduced GFP signal, while HY5-S36D showed accumulated GFP signal similar to the wild-type HY5-GFP at high ambient temperature (Fig. 4d, e). Although GFP fusion may affect the expression and stability of a fusion protein, overall these data suggest that the phosphorylation of HY5 is necessary to stabilize HY5 in the root to promote primary root elongation at high ambient temperature.

**SPAs are necessary for root thermomorphogenesis.** Recently, we showed that SPA proteins are necessary for the thermomorphogenesis phenotype of the shoot[24]. To examine whether the SPAs are also involved in regulating root thermomorphogenesis, we tested high ambient temperature-mediated root elongation phenotype using *spa* higher order mutants (*spa123* and *spa1234*, *spaQ*) (Fig. 5a, b). Results show that the thermo-induced root elongation was attenuated in the *spa123* and absent in *spaQ* mutant compared to WT. The primary root length was even shorter for *spaQ* at 28 °C vs 22 °C, suggesting that all four *SPA* genes are necessary for root elongation.

Lee et al. also showed that SPA1 directly phosphorylates and stabilizes PIF4 to promote hypocotyl and petiole elongation[24]. To test whether the kinase activity of SPA1 is necessary for root thermomorphogenesis, we grew previously described transgenic plants expressing the WT SPA1 (35S:LUC-SPA1/*spaQ*) or mutant SPA1 (35S:LUC-mSPA1/*spaQ*) affecting its kinase activity in the *spaQ* background[46]. Examination of the primary root length phenotype showed that the WT SPA1 can largely rescue the root elongation phenotype of *spaQ*. However, the mutant SPA1 affecting its kinase activity failed to rescue the root elongation phenotype of *spaQ* at high ambient temperature (Fig. 5a–c), suggesting that the kinase activity of SPA1 is necessary for root thermomorphogenesis.

To test more directly whether the kinase activity of SPA1 is regulated by temperature, we performed in vitro kinase assays using WT HY5 as a substrate at normal (22 °C) and high ambient temperature (28 °C). Strikingly, the results show that the kinase activity of SPA1 is stimulated >2-fold at 28 °C (Supplementary Fig. 11). Given that HY5 is stabilized by SPA1-mediated phosphorylation of serine 36 of HY5 (Fig. 4d, e; Supplementary Fig. 10)[44], these data suggest that SPA1 might phosphorylate HY5 more at high ambient temperature to stabilize it. This is also consistent with the SPA1-mediated phosphorylation and stabilization of PIF4[24].

We also examined the hypocotyl thickness phenotypes of the above lines in response to high ambient temperature (Supplementary Fig. 12). Results show that the thickness of *spaQ* hypocotyl at 22 °C was higher than that of WT at 28 °C. Moreover, *spaQ* hypocotyl became much thicker at 28 °C compared to 22 °C possibly due to reduced elongation. To test whether the kinase activity is necessary for the hypocotyl thickness phenotype, we also analyzed the transgenic plants expressing the WT SPA1 (35S:LUC-SPA1/*spaQ*) or mutant SPA1 (35S:LUC-mSPA1/*spaQ*) affecting its kinase activity in the *spaQ* background as described above. The data show that the WT SPA1 largely rescued the hypocotyl thickness phenotype of the *spaQ*. By contrast, the mutant SPA1 slightly reduced the hypocotyl thickness of the *spaQ* mutant in response to high ambient temperature. These data suggest that the kinase activity of SPA1 is

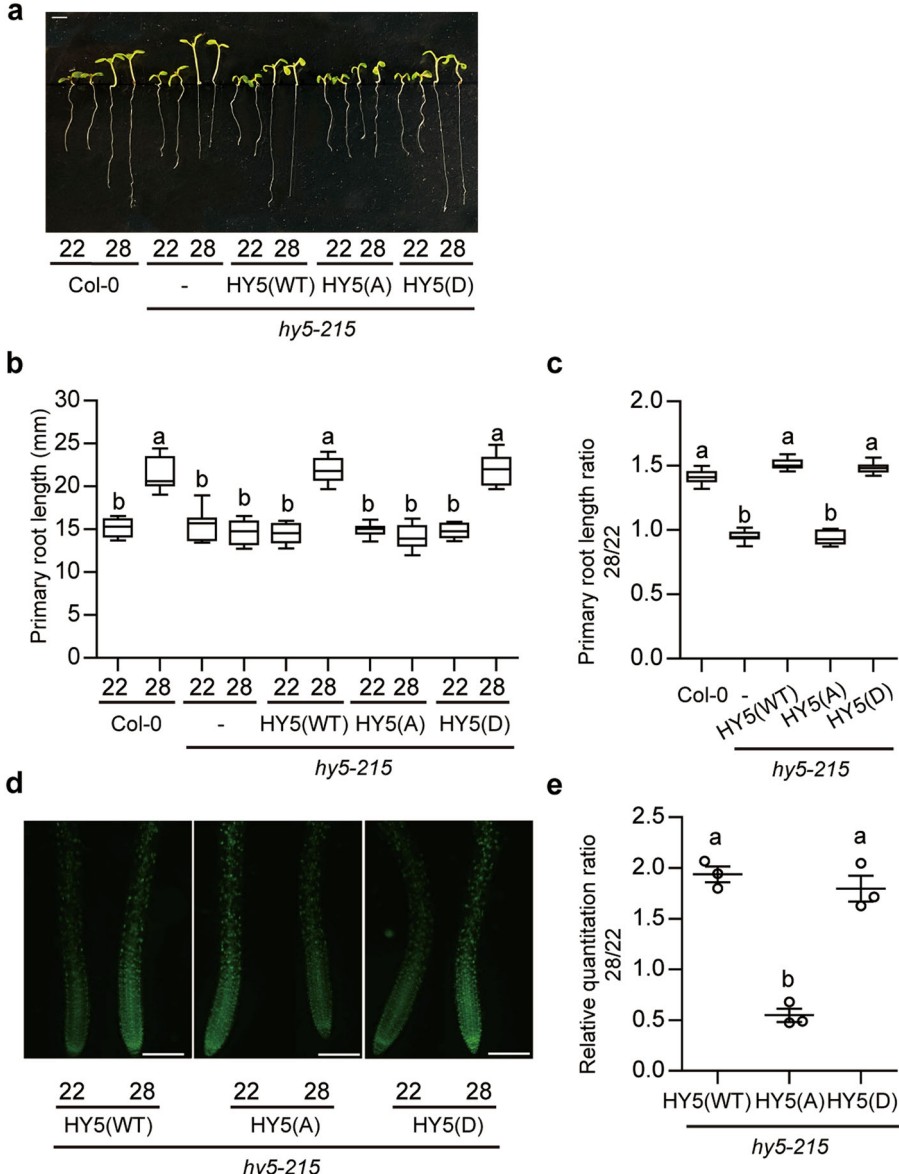

**Fig. 4 Phosphorylation of HY5 is necessary for stabilization and root thermomorphogenesis. a** Photograph showing seedling phenotypes of three different types of HY5 overexpression lines grown in 22 °C and 28 °C. Seedlings were grown for two days in continuous white light at 22 °C and then either kept at 22 °C or transferred to 28 °C for additional 4 h before being photographed. The scale bar represents 2 mm. **b**, **c** Bar graphs show the primary root lengths (**b**) or the root length ratio (**c**) for the seedlings grown under conditions described in (**c**). For the length ratio 28/22, the primary root length at 28 °C is divided by that at 22 °C. The letters a–c indicate statistically significant differences between means of primary root lengths ($n = 12$, $P < 0.05$) based on one-way ANOVA analyses with Tukey's HSD test. For the box and whisker plots in **b**, **c**, the boxes represent from the 25th to the 75th percentile and the bars equal the median values. The source data underlying the root measurements (**b**, **c**) are provided in the Source data file. **d** Fluorescence images of roots using three types of HY5-GFP overexpression lines with the same exposure time ($t = 300$ ms). The scale bar represents 200 μm. **e** Bar graph shows the relative amount of HY5-GFP signal at 28 °C divided by that at 22 °C with three repeats. Error bars indicate SD ($n = 3$). The letters a, b indicate statistically significant differences between means of primary root lengths ($n = 3$, $P < 0.05$) based on one-way ANOVA analyses with Tukey's HSD test.

also necessary for regulating the hypocotyl thickness in response to high ambient temperature.

**PIF4 and HY5 are spatially regulated**. Because PIF4 and HY5 appear to regulate thermomorphogenesis largely in a tissue-specific manner, we tested the expression of these genes using whole seedlings, shoot and root tissues grown at 22 °C and then transferred to 28 °C for 4 h. qPCR analyses showed that the expression of *PIF4* is upregulated only in the shoot tissues with similar level in the root (Fig. 6a). Strikingly, the expression of *HY5* showed specific upregulation only in the root tissue (Fig. 6b),

with similar expression in the shoot and total seedlings in response to high ambient temperature. Although the growth conditions are slightly different, these expression data are largely consistent with the expression of *PIF4* and *HY5* in publicly available datasets (Supplementary Fig. 13).

To test whether the expression pattern of these genes reflects the protein level in these tissues, we also performed immunoblot analyses of PIF4 and HY5 using shoot and root tissue samples grown under the same conditions. Results show that PIF4 is more abundant in the shoot, while the HY5 level is upregulated specifically in the root in response to high ambient temperature

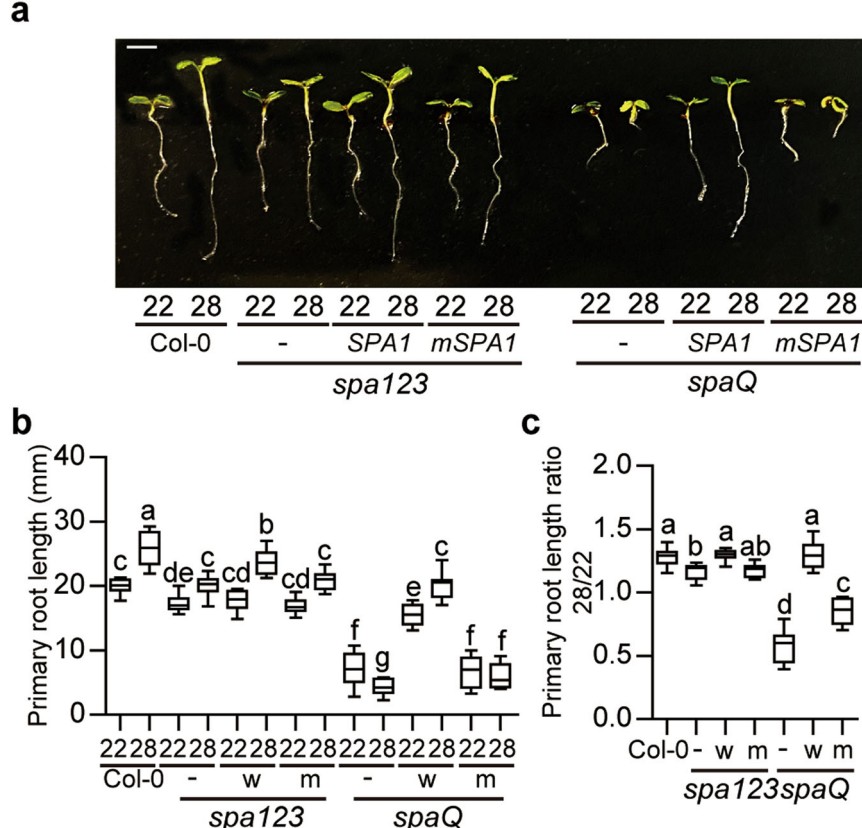

**Fig. 5 SPAs are necessary for root thermomorphogenesis. a** Photograph shows the seedling phenotypes of Col-0, *spa123*, *35S:LUC-SPA1/spa123*, and *35S: LUC-mSPA1/spa123*, *spaQ*, *35S:LUC-SPA1/spaQ*, and *35S: LUC-mSPA1/spaQ* at 22 °C and 28 °C. Three *spa* background plants are indicated with—for *spa123* or *spaQ*, *SPA1* for *35S:LUC-SPA1* overexpression line, and *mSPA1* for *35S:LUC-mSPA1* overexpression line, respectively. Seedlings were grown for two days in continuous white light at 22 °C and then either kept at 22 °C or transferred to 28 °C for additional 5 days before being photographed. The scale bar represents 2 mm. **b**, **c** Bar graph shows the primary root lengths (**b**) and the root length ratio (**c**) for seedlings grown under conditions described in (**a**). w, wild-type SPA1 and m, mSPA1, respectively. For the length ratio 28/22, the primary root length at 28 °C was divided by that at 22 °C. The letters a–f indicate statistically significant differences between means of primary root lengths ($n = 15$, $P < 0.05$) based on one-way ANOVA analyses with Tukey's HSD test. For the box and whisker plots in **b**, **c**, the boxes represent from the 25th to the 75th percentile and the bars equal the median values. The source data underlying the root measurements (**b**, **c**) are provided in the Source data file.

(Fig. 6c). In fact, HY5 level is slightly reduced in the shoot tissue in response to high ambient temperature (Supplementary Fig. 14), while PIF4 remains unchanged in the root tissue. To examine whether the phenotypic difference of root thermomorphogenesis under LD and SD reflects the protein levels, immunoblot of HY5 was performed using samples grown under LD and SD conditions (Supplementary Fig. 15). The results show that while the basal level of HY5 in the root was higher under LD compared to SD conditions, HY5 was more stabilized by high ambient temperature under LD compared to SD conditions. While the lack of stabilization of HY5 at 28 °C compared to 22 °C under SD vs LD may not solely explain the differences in root thermomorphogenesis phenotype under these conditions, these data suggest that proper level of HY5 in the root might be essential for triggering root thermomorphogenesis. Overall, these tissue-specific gene expression and protein abundance of PIF4 and HY5 might explain why plants employ these two transcription factors to regulate thermomorphogenesis in a tissue-specific manner.

## Discussion

Global warming caused by climate change has a profound impact on crop productivity[1,47]. However, our understanding of the impact of high ambient temperature is mostly based on the above ground tissue responses[2,3]. Our comprehensive phenotypic, genetic, biochemical and genomic analyses show that plants respond to high ambient temperature in a tissue-specific manner and employ distinct transcription factors to regulate spatial thermomorphogenesis.

Root system architecture plays a crucial role in crop productivity[28,29,48] and high ambient temperature affects both primary and lateral root growth[26,27,30]. The phenotypic analyses show that seedlings display a general growth promotion including elongation of hypocotyls, petioles and roots, and thickening of hypocotyls in response to high ambient temperature. Moreover, day-length profoundly affects growth responses with stronger impact on root elongation under SD conditions. This is in sharp contrast where hypocotyl elongates more under SD compared to LD or constant light conditions. In addition, global gene expression analyses show that root and shoot respond to high ambient temperature largely differently with a small set of overlapping genes. These data are consistent with the previous reports that plants respond to high ambient temperature in a tissue-specific manner[30,34].

Previously, it was shown that PIF4 acts as a main hub transcription factor regulating plant responses to high ambient temperature[9]. PIF4 plays a minor, if any role, in root responses to high ambient temperature[30]. Our genetic analyses suggest that while PIF4 acts a hub transcription factor regulating the above ground tissue-responses to high ambient temperature, HY5 acts as a key transcription factor regulating root responses to high ambient temperature. A recent report concluded that PIF4 is

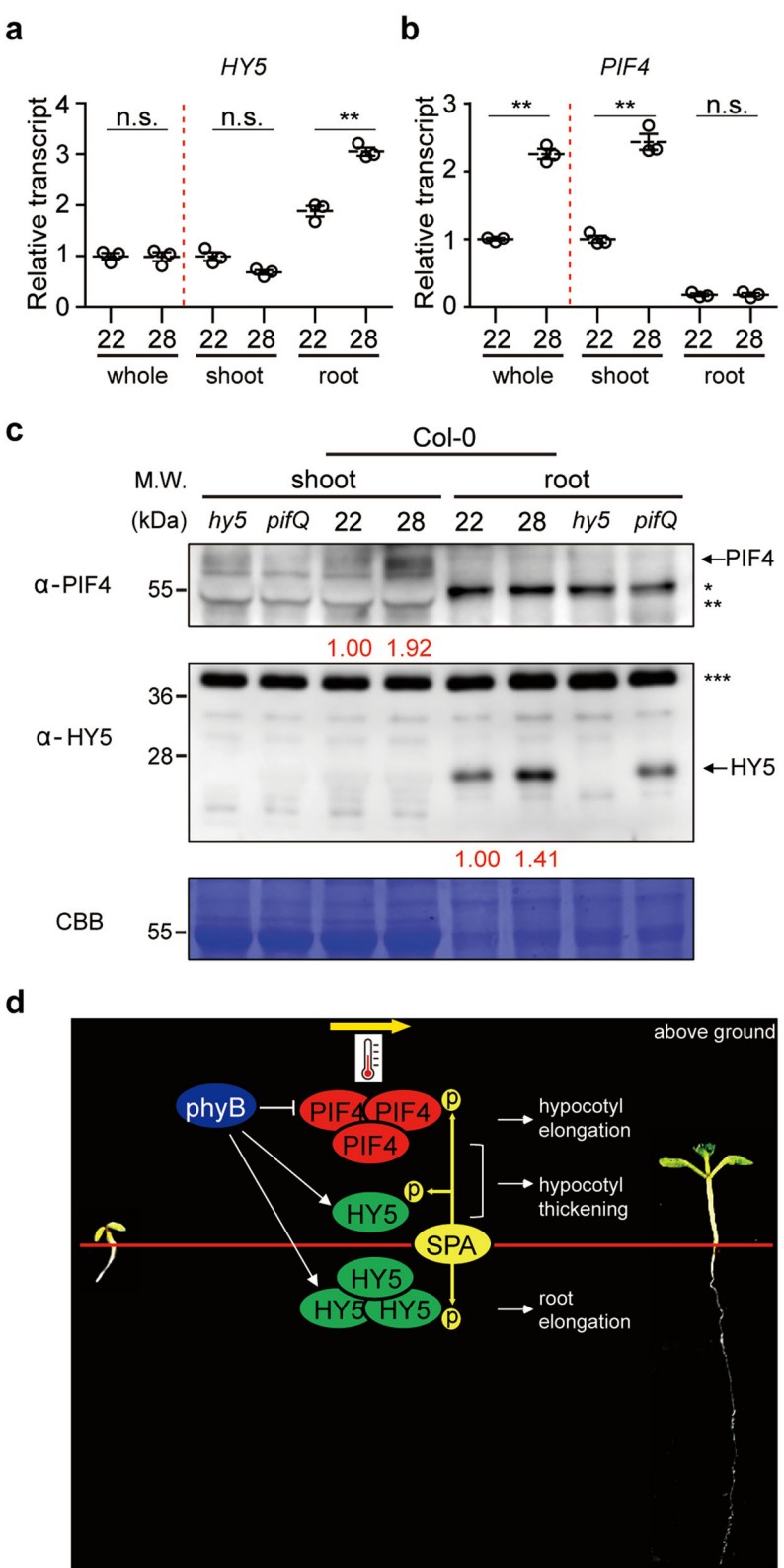

involved in root thermomorphogenesis based on *PIF4* over-expression line[34]. However, our genetic analyses data using *pif4-2* and *pifQ* mutants are consistent with a previous conclusion that PIF4 does not play a role in root thermomorphogenesis[30]. These data are also consistent with the tissue-specific expression and protein abundance of PIF4 and HY5 in shoot and root, respectively.

The genomic analyses suggest that HY5 directly/indirectly regulates a large number of genes in both shoot and root tissues consistent with its role as previously shown[37,42,43,49]. However, the shoot and root-specific transcriptomic changes are largely distinct with a small overlapping set, highlighting a distinct role of HY5 in regulating the above and below-ground tissue responses to high ambient temperature. A more in-depth gene

**Fig. 6 The expression and stability of PIF4 and HY5 are spatially regulated. a, b** RT-qPCR analysis of *HY5* and *PIF4*. RT-qPCR samples were from Col-0 whole seedlings, shoot or root grown for 5 days at 22 °C and then either kept at 22 °C or transferred to 28 °C for 4 h. Three biological repeats were performed. Relative gene expression levels were normalized using the expression levels of ACT7 as well as the expression levels of shoot. Asterisks indicate statistically significant difference using two-sided Student's *t*-test; **p < 0.01. P values for *HY5* root 22 vs 28: 0.0009, *PIF4* whole seedlings: 0.0005, *PIF4* shoot: 0.0004, respectively. Error bars indicate SD (*n* = 3). **c** Western blots show the level of native PIF4 and HY5 in the shoot and root samples separated from whole seedlings grown for 5 days at 22 °C and then either kept at 22 °C or transferred to 28 °C for 4 h. *hy5-215* and *pifQ* were used as negative controls. * and ** indicate root and shoot specific cross-reacting bands. *** indicate cross-reacting band in HY5 blot. Red numbers indicate the relative values for PIF4 or HY5 band divided by the cross-reacting band. Two independent experiments were repeated with similar results. **d** Simplified scheme showing spatial regulation of thermomorphogenesis by distinct transcription factor-kinase module. PIF4 and HY5 are main hub transcription factors in the shoot and root thermomorphogenesis, respectively, while PIF4 and HY5 both contribute to hypocotyl thickness. The abundance and/or activity of PIF4 and HY5 are regulated by SPA-mediated phosphorylation. As a thermosensor, phyB directly inhibits PIF4 and indirectly stabilizes HY5 to coordinately regulate shoot and root thermomorphogenesis.

expressing analyses using both RNAseq and qRT-PCR showed that HY5 directly regulated the expression of both auxin and BR pathway genes to regulate root thermomorphogenesis consistent with the previous reports[30,34]. HY5 protein has been shown to transport from shoot to root[50]. Using shoot-specific expression of HY5 (*pCAB3:DOF-HY5*), Gaillochet et al. also showed that the shoot derived HY5 partially rescues the root thermomorphogenesis phenotype of *hy5*[34]. However, our data show that HY5 gene expression and protein stability are both stimulated by high ambient temperature in a root-specific manner, suggesting that the root-derived HY5 might play a large role in regulating root thermomorphogenesis.

Recently, we have shown that SPAs play a crucial role in regulating thermomorphogenesis by controlling the phyB-PIF4 module[24]. Moreover, SPA1 directly phosphorylates and stabilizes PIF4 to regulate thermomorphogenesis. In addition, SPAs directly phosphorylate and stabilize HY5[44]. Unphosphorylated HY5 is more active, but a preferred substrate for COP1-mediated degradation as previously shown[44,51]. The phenotypic data show that SPAs regulate both shoot and root thermomorphogenesis, and the kinase activity of SPA1 is not only necessary for the biological function of SPA1, but also stimulated by high ambient temperature. Thus, SPAs might directly phosphorylate both PIF4 and HY5 to regulate the above- and below-ground tissue responses to high ambient temperature (Fig. 6d).

While HY5 regulates root thermomorphogenesis, our data show that its role is not only restricted to the root tissue. In fact, both HY5 and PIFs are involved in regulating hypocotyl thickness in response to high ambient temperature in an interdependent manner. Moreover, SPA kinase activity is also involved in regulating hypocotyl thickness. While PIFs and HY5 have been shown to regulate light and temperature responses in an antagonistic manner[36], PIF1/PIF3 and HY5/HYH have been shown to interact with each other and coordinately regulate reactive oxygen species signaling in Arabidopsis[52]. Thus, PIFs-HY5-SPA module might regulate hypocotyl thickness in the above ground tissue. Further studies are necessary to understand how this regulatory module controls hypocotyl thickness in response to high ambient temperature.

In summary, the data presented here suggest that plants employ distinct transcription factors for regulating thermomorphogenesis in a tissue-specific manner. While PIF4-SPA module plays a pivotal role in hypocotyl elongation[24], HY5-SPA module plays a crucial role in regulating root thermomorphogenesis (Fig. 6d). In addition, PIFs-HY5-SPA module regulates hypocotyl thickness in response to high ambient temperature. Moreover, a number of factors including, Cryptochrome 1 (Cry1), ELF3, TOC1, HMR, BBX18/23, TCP, FCA, COP1-DET1-HY5 module, and GI-RGA-PIF4 module converges on PIF4 to control its expression and/or abundance to regulate shoot thermomorphogenesis[13–24]. Further studies are necessary to

understand if any of these factors or others also regulate root thermomorphogenesis in concert with HY5.

## Methods

**Plant materials, growth conditions, and phenotypic analyses**. All the seeds used in this study are from Col-0 ecotype of *Arabidopsis thaliana*. Seeds were surface-sterilized and plated on Murashige and Skoog (MS) media without any sucrose. Seeds were stratified for 3 days at 4 °C, and then were placed in 22 °C for 2 days and transferred to 22 °C or 28 °C for additional 4 days or as indicated in figure legends. For soil-grown root elongation experiments, seeds were germinated on MS plates for 3 days and then the seedlings were transferred to soil and grown for additional 5 days at either 22 °C or 28 °C under LD conditions. White light intensity was 30 μmol m$^{-2}$ s$^{-1}$. Hypocotyl length, hypocotyl thickness, and primary root length with 10–15 seedlings were measured using ImageJ software and analyzed using one-way ANOVA with Tukey's HSD test.

**Protein extraction and western blot analyses**. Total protein extracts were made from 50 seedlings for each shoot or root sample using 50 μL urea extraction buffer, consisting of 8 M urea, 0.35 M Tris-Cl pH 7.5, and 1× protease inhibitor cocktail. After boiling with 6× SDS sample buffer, samples were centrifuged at 20,200 × g for 15 min and loaded into SDS-PAGE gels. Separated proteins were transferred onto PVDF membrane (Millipore) and then immunoblotted using anti-GFP (Abcam, ab290, dilution 1:5000), anti-PIF4 (Agrisera, AS16 3955, dilution 1:2000), or anti-HY5 (Abiocode, R1245-2, dilution 1:3000) antibodies. For the secondary antibodies, anti-mouse (Abcam, ab131368, dilution 1:5000), anti-rabbit (KPL, 95059-086, dilution 1:50,000), or anti-goat (Agrisera, AS09 605, dilution 1:5000) were used. Coomassie blue staining (CBB), anti-Tubulin (Sigma-Aldrich, T5168, dilution 1:5000), or anti-RPT5 (Enzo Life Sciences, BML-PW8770-0025 dilution 1:5000), was used for loading control. SuperSignal West Atto Chemiluminescent substrate (ThermoFisher Scientific, A38556) was used for visualizing secondary HRP bound antibodies.

**RNA extraction, cDNA synthesis, and qRT-PCR**. Whole seedlings, shoot and root tissues from 6-day-old white light-grown seedlings were used with three independent biological replicates (*n* = 3) for RNA extraction. Seedlings were grown at 22 °C under continuous white light for 6 days and then were either kept at 22 °C or transferred to 28 °C for additional 4 h under continuous white light before samples were harvested. Plant RNA purification kit from Sigma-Aldrich were used for total RNA extraction following the manufacturer's protocols. For cDNA synthesis, 1 μg of total RNA was used for reverse transcription with M-MLV Reverse Transcriptase (Thermo Fisher Scientific). SYBR Green PCR master mix (Thermo Fisher Scientific) and gene-specific oligonucleotides were used to conduct qPCR analyses. Finally, relative transcription level was calculated using $2^{\Delta Ct}$ with *ACT7* as a control.

**RNA-seq analyses**. 3'Tag-Seq method was used in this study for RNA-seq analysis[53]. FastQC was used to access raw read quality (www.bioinformatics.babraham.ac.uk/projects/fastqc/). The raw reads were aligned to the Arabidopsis genome using HISAT2[54]. Data from TAIR10 (www.arabidopsis.org/) was used as the annotation of the Arabidopsis genome. HTseq module was used for counting reads[55] (htseq.readthedocs.io/en/ master/). Differentially expressed genes in WT and *hy5* were identified using the EdgeR[56]. Cutoff and adjusted P value (FDR) for the differential gene expression were setup with ≥2-fold and ≤0.05, respectively. Venn diagrams were generated using the website (http://bioinformatics.psb.ugent.be/webtools/Venn/). Heatmap was generated using Morpheus (https://software.broadinstitute.org/morpheus/). For the heatmap, hierarchical clustering was used with one minus cosine similarity metric combined with average linkage method. Also, GO enrichment analyses were performed using GENE ONTOLOGY (http://geneontology.org/). GO bar graphs were generated based on the result of the significantly enriched terms with the lowest P value and FDR (≤0.05) in GO terms.

Raw data and processed data for RNA-Seq in Col-0 and *hy5-215* can be accessed from the Gene Expression Omnibus database under accession number GSE158939.

**Chromatin immunoprecipitation (ChIP) assays**. Chromatin Immunoprecipitation (ChIP) assays were conducted as described previously[24,46]. For ChIP assay samples, 6-day-old seedlings of 35S:HY5-GFP in *hy5-215* background were transferred to 22 °C or 28 °C for 24 h under continuous white light and then harvested. In brief, samples were crosslinked using 1% formaldehyde under vacuum for 15 min and then 2 M glycine was added for additional 5 min for quenching. Samples were washed with distilled water for five times and ground with mortar and pestle using liquid nitrogen. Ground samples were placed into 1.5 mL microtube with nuclei isolation buffer (0.25 M sucrose, 15 mM PIPES, 5 mM $MgCl_2$, 60 mM KCl, 15 mM NaCl, 1% Triton X-100, 2 mM PMSF, 1× protease inhibitor cocktail) for 15 min and then centrifuged at $20,200 \times g$ for 10 min at 4 °C. Pellets were resuspended with 1 mL lysis buffer (50 mM HEPES pH 7.5, 150 mM NaCl, 1 mM EDTA, 1% Triton X-100, 0.1% SDS, 2 mM PMSF, 1× protease inhibitor cocktail). Then, sonication of chromatin pellet was performed using digital sonifier (Branson, Model# PC BD 400 W 20 kHz). ChIP grade anti-GFP (Abcam, ab6556) coupled to dynabeads were used for immunoprecipitation. After washing steps including low salt wash buffer, high salt wash buffer, LiCl wash buffer, and TE buffer, samples were eluted with elution buffer. By adding NaCl with final 0.2 M, samples were incubated overnight at 65 °C for reverse crosslinking. Finally, PCR purification kit (QIAGEN) was used for DNA purification after proteinase K treatment for an hour. Samples without IP were used as input DNA. Enrichment (% of input) was calculated from each sample relative to their corresponding input.

**Microscopy**. For light microscopy, seedlings were placed in 22 °C for 2 days and transferred to 22 °C or 28 °C for additional 4 days. Samples were inoculated in 100% EtOH overnight and light microscopy was conducted. For fluorescence image, seedlings were placed in 22 °C for 5 days and transferred to 22 °C or 28 °C for additional 4 h in white light condition. Root images of transgenic seedlings were obtained using Olympus BX53 with 300 ms exposure time.

**Reporting summary**. Further information on research design is available in the Nature Research Reporting Summary linked to this article.

## Data availability
RNA sequencing data were deposited into the Gene Expression Omnibus database (accession number GSE158939). Arabidopsis mutants and transgenic lines, as well as plasmids and antibodies generated during the current study are available from the corresponding author upon reasonable request. Source data are provided with this paper.

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

## Acknowledgements

We thank members of the Huq laboratory for critical reading of the manuscript. We thank Drs. Mijeong Kim and Taeyoung Lee for comments analyzing RNA-seq. This work was supported by grants from the National Science Foundation (MCB-2014408) and National Institute of Health (NIH) (GM-114297) to E.H., and Integrative Biology (IB) Research Fellowship grant from the University of Texas at Austin to S.L. The authors acknowledge the Texas Advanced Computing Center (TACC) at The University of Texas at Austin for providing High Performance Computing, visualization, and database resources that have contributed to the research results reported in this paper.

## Author contributions

S.L., W.W., and E.H. conceived the study and designed the experiments. S.L. and W.W. carried out the experiments. S.L., W.W., and E.H. analyzed the data. S.L. and E.H. wrote the paper.

## Competing interests

The authors declare no competing interests.
