## [Peer Review File · Nature Communications]

REVIEWER COMMENTS

Reviewer #1 (Remarks to the Author):

Spatial regulation of thermomorphogenesis by HY5 and PIF4 in Arabidopsis

The manuscript by Lee et al. purport to show that temperature dependent root elongation is controlled by the transcription factor HY5 through a spatially specialised mechanism from that of the above ground parts, which is controlled by PIF4.

I was very excited initially with the concept, which could have offered a mechanistic insight into the recently unravelled role of HY5 in root thermal responses. I was however let down by the manuscript on several fronts. The manuscript suffers from major conceptual flaws and technical inadequacies as discussed below. To this reviewer, the analyses and interpretations appear less than rigorous. Major issues include exaggerated and not well-supported claims and a rather casual manner of approach. Seedling images are of poor quality. The manuscript, it appears, is put together without the necessary robustness and important details in relation to the reagents, experimental rationale and data underpinning major arguments are left wanting.

The authors concentrate their studies on constant light (CL) grown plants. While they have provided some data on long-day and short-day grown plants. It would be informative to be essential to check HY5 protein in root and shoot at these different conditions.

Given that HY5 is predominantly regulated by the COP1 and DET1 through a mechanism that involves the degradation of the HY5 protein in dark. Under natural conditions, roots growing in soil are not exposed to light, a condition conducive for the degradation of HY5. The experimental condition of continuous light (CL) for seedlings growing on agar plates therefore is conceptually flawed for assessing root function. Moreover, HY5 has been shown to be a shoot-to-root mobile protein that controls root growth through in a light-dependent manner.

The authors need to be careful of their mention of spatial separation of mechanisms in root and shoot.

The authors claim of "Comprehensive analysis of spatial thermomorphogenesis at seedlings stage" is exaggerated and couldn't be far from reality. There are a numerous conceptual and technical flaws related to these experiments.

While it is absolutely appropriate and necessary to bring up the role of PhyB in thermosensing, the use of the term 'thermosensor mutant Phyb' is inaccurate. Indeed, PhyB has been shown to play a role in temperature perception, but the loss of function PhyB should not be called a 'thermosensor mutant'.

The authors have very casually used mutant names. They should specifically mention in the text the alleles they have used, and this should be clearly shown in figures. No information of the reagents is given elsewhere either!

Beyond the pictorial representation in Figures, the authors have not provided analysed data from gene expression analyses (RNA-seq) shown in Figure 3.

Based on their RNA-seq data (which this reviewer could not verify due to lack of supporting data) the authors have selected IAA19, SAUR40 and SAUR77 genes involved in auxin signalling for further analyses. The authors argue that these genes are HY5 targets by referring to a published data. What would be needed is to analyse HY5 binding to these and other key genes in the root and shoot at the experimental conditions.

Have the authors performed experiments for RNA-seq and the qRT-PCR experiment shown in Fig S7? Why do they see different results for BR-related genes? The authors show that unlike their RNA-seq data, expression of BR metabolic genes is temperature responsive. This temperature responsiveness was not observed in *hy5* mutant suggesting that HY5 may be essential. Why the authors did not include 35S:HY5 in this analysis to check if it is sufficient?

The authors perform ChIP analysis on BAS1 and SOB7, not CPD, for studying HY5 binding. The rationale, it seems, for not including CPD in this analysis was that they could not detect G-box element or its variants in the 2kb upstream sequence. Why limit at 2 kb? There have been examples of functional CREs downstream the TSS, within the gene body and even 3'.

In which tissue was the HY5 ChIP experiment performed? It would have been meaningful if these

analyses were done under the experimental conditions with different temperatures. Why was the HY5 western blot shown in Fig 4A and Fig S9 also not performed for root and shoot separately instead of only whole seedlings? The authors analyse expression of the HY5 transgene in shoot and root separately (Fig S8). This, along with other such discrepancies, makes this reviewer sceptical of the robustness of the authors analyses and wonder if they pick and present what is more convenient rather than providing a rigorous analysis! The argument of "Phosphorylation of HY5 is necessary....." is not sufficiently supported. At best they have shown that HY5-S35 has a modest effect on temperature response as shown in Fig 4D. In addition, the phosphomimic HY5-S35D did not show any phenotype. Therefore, the question of whether phosphorylation of HY5 is affected by temperature per se is not directly addressed. Moreover Fig. 4C, D and E should have used the *hy5* mutant as a control. The conclusions derived from *spaQ* mutant data (Fig 5) is not convincing as it seems *spaQ* is severely growth-defective, particularly very sensitive to 28°C. This suggests the important role SPAs play in general growth and development. Therefore, what is interpreted as a reduced or lack of root temperature responsiveness could well be a reflection of the defective development/growth in general rather than specific response of roots. The immunoblot data shown in Fig 6 is unclear due to what appears to be non-specific hybridizations and the lack of proper controls to unambiguously provide confidence in the data. There is no size marker or a negative control reagent (such as *pif4* or *hy5* mutants). Nor is there any loading control. These issues need to be clarified. Finally, the model proposed by the authors and therefore the schematic representation in Fig 6 is not fully supported. A clear explanation of how antagonistic transcription factors PIF4 and HY5 spatially uncouple their activities as proposed here to control temperature responsiveness in shoot and root respectively is not amply supported by experimental evidence. In summary, the data presented here and the discussion thereafter, although interesting, are preliminary in its current form. The role of HY5 in root elongation and particularly temperature response is known. A rigorous follow up of what appears to be preliminary and less than rigorous analysis, to this reviewer, here could make this an interesting work that will unravel the molecular mechanism by which temperature controls root growth. It is important, however, to carefully address the conceptual and technical flaws.

Reviewer #2 (Remarks to the Author):

Authors for this manuscript demonstrate that two distinct transcription factors regulate thermo responses spatially. PIF4 is the key player regulating shoot responses to high ambient temperature. HY5 is the one regulating root thermo responses. In addition, both HY5 and PIFs are involved in regulating hypocotyl thickness in response to high ambient temperature. Their tissue-specific accumulation requires SPA1 or all four SPAs. They also show that root and shoot respond to high ambient temperature differently with only a small set of overlapping.

Major

1. Figures 4A and S9, separate root and shoot blots are needed to reflect possibly a large difference. In addition, normalization is preferred to a protein band through immunoblotting. The authors should also acknowledge that GFP fusion may not fully reflect the accumulation of a fusion protein since GFP tends to make a fusion protein much stable.
2. In Figure 6C, better quality immunoblot of PIF4 is required and new HY5 immunoblot if possible. To show the specificity of the antibody, protein preparations from a *pif4* mutant and a *hy5* mutant should be loaded side by side.
3. Are there two bands for PIF4 in shoot tissue and two bands of HY5 in root tissue? The fast migration band may be the phosphorylated band.
4. For Figure 6D, authors are encouraged to add the function of HY5 in thermo-induced hypocotyl thickness to the model. How is HY5 involved in thermo-induced hypocotyl thickness while its transcription is mostly thermo-induced in the root tissue? Its protein accumulation follows a similar pattern. Is the background level of HY5 sufficient to trigger the response?
5. Both PIF4 and HY5 are involved in thermo-induced hypocotyl thickness. This is a very specific phenotype. Authors could discuss how they likely work together.

Minor

1. On page 8, change three different HY5 to three different HY5 forms.
2. In Figure 5 legend, the root length ration should be corrected to the root length ratio.

Reviewer #3 (Remarks to the Author):

This manuscript by Lee et al. reports the interesting spatial regulation of Arabidopsis thermomorphogenesis by HY5 and PIF4. PIF4 has been shown to act as a hub transcription factor to regulate the shoot responses to high ambient temperature, while in this study, the authors showed that HY5 acts as a key transcription factor to regulate root responses to high ambient temperature. The authors further showed that HY5 phosphorylation is required to stabilize HY5 in the root to promote primary root elongation at high ambient temperature, and that the kinase activity of SPA1 is necessary for root thermomorphogenesis. Thus, PIF4-SPA module and HY5-SPA module play pivotal roles in shoot and root, respectively, in plant response to high ambient temperature. Together, the data in this manuscript suggest that plants employ distinct combinations of the SPA-PIF4-HY5 module to regulate tissue-specific thermomorphogenesis. In general, the data presented in this manuscript are novel and interesting, and contribute to a better understanding of high ambient temperature-induced architecture changes of sessile plants. Here are my suggestions to improve this manuscript.

1. Fig 2 shows that phyB regulates both above and below-ground tissues at high ambient temperature. Since phyB has been shown to be a thermosensor, it will be helpful to reflect the role of phyB in the model (shown in Figure 6D).
2. Since the kinase activity of SPAs are essential for shoot and root thermomorphogenesis, how the protein levels or kinase activities of SPAs are regulated by high ambient temperatures? Also, it will be interesting to investigate whether HY5 phosphorylation is regulated by high temperature.
3. Fig 6A and C show that the elevated HY5 protein level in root at high ambient temperature may be caused by upregulation of its gene expression. So how the authors define the contribution of SPA1-mediated phosphorylation in regulating HY5 protein abundance at high ambient temperature?

Minor points:

1. Fig 3A: 2209 of HY5 regulated genes in total does not equal 665+418.
2. Statistical analyses are lacking in Fig 4B and 4G.
3. P4: "Martins et al., 2017" should be replaced by the number "29".
4. P5, first paragraph: Fig 4A, B and C should be appropriately cited.
5. P6, third paragraph: "Fig. S6B" should be Fig. S6A.
6. Protein markers are suggested to be shown for Fig 6C.

Jigang Li

Response to Reviewer's Comments

Reviewer #1 (Remarks to the Author):

Spatial regulation of thermomorphogenesis by HY5 and PIF4 in Arabidopsis

The manuscript by Lee et al. purport to show that temperature dependent root elongation is controlled by the transcription factor HY5 through a spatially specialised mechanism from that of the above ground parts, which is controlled by PIF4.

I was very excited initially with the concept, which could have offered a mechanistic insight into the recently unravelled role of HY5 in root thermal responses. I was however let down by the manuscript on several fronts. The manuscript suffers from major conceptual flaws and technical inadequacies as discussed below. To this reviewer, the analyses and interpretations appear less than rigorous. Major issues include exaggerated and not well-supported claims and a rather casual manner of approach. Seedling images are of poor quality. The manuscript, it appears, is put together without the necessary robustness and important details in relation to the reagents, experimental rationale and data underpinning major arguments are left wanting.

The authors concentrate their studies on constant light (CL) grown plants. While they have provided some data on long-day and short-day grown plants. It would be informative to be essential to check HY5 protein in root and shoot at these different conditions.

Author response: This is an excellent suggestion. We have now added native HY5 protein level in long-day (LD) and short-day (SD) grown plants. (Fig.S15) The result shows that HY5 accumulates more in the root samples under LD condition compared to SD conditions. In contrast, HY5 accumulates more in the shoot samples under SD than in LD conditions. These data suggest that a certain level of HY5 is needed to trigger root thermomorphogenesis.

Given that HY5 is predominantly regulated by the COP1 and DET1 through a mechanism that involves the degradation of the HY5 protein in dark. Under natural conditions, roots growing in soil are not exposed to light, a condition conducive for the degradation of HY5. The experimental condition of continuous light (CL) for seedlings growing on agar plates therefore is conceptually flawed for assessing root function. Moreover, HY5 has been shown to be a shoot-to-root mobile protein that controls root growth through in a light-dependent manner.

Author response: This is an excellent suggestion. We have now added the data showing seedlings grown in soil conditions (Fig. S2). Similar to the petri dish-grown seedlings under CL and LD conditions, Col-0 and *pifQ* mutant showed elongated primary root phenotype at high ambient temperature in soil, while *hy5* mutant did not respond to high ambient temperature. These data support that HY5 is essential for root thermomorphogenesis in both artificial and natural conditions.

The authors need to be careful of their mention of spatial separation of mechanisms in root and shoot.

The authors claim of “Comprehensive analysis of spatial thermomorphogenesis at seedlings stage” is exaggerated and couldn’t be far from reality. There are a numerous conceptual and technical flaws related to these experiments.

Author response: We have removed the word “comprehensive”. We have also added new data and hope that we have addressed all the technical issues raised by this reviewer by performing the natural soil conditions and additional supporting experiments under LD and SD conditions.

While it is absolutely appropriate and necessary to bring up the role of PhyB in thermosensing, the use of the term ‘thermosensor mutant Phyb’ is inaccurate. Indeed, PhyB has been shown to play a role in temperature perception, but the loss of function PhyB should not be called a ‘thermosensor mutant’.

Author response: This has been corrected and we modified the text to reflect these changes.

The authors have very casually used mutant names. They should specifically mention in the text the alleles they have used, and this should be clearly shown in figures. No information of the reagents is given elsewhere either!

Author response: These have been corrected. We modified *phyb* to *phyb-9*, *pif4* to *pif4-2*, and *hy5* to *hy5-215* in the text and figures. Native PIF4 and HY5 antibody information are also added in the method section anti-PIF4 (Agrisera, AS16 3955), or anti-HY5 (Abiocode, R1245-2).

Beyond the pictorial representation in Figures, the authors have not provided analysed data from gene expression analyses (RNA-seq) shown in Figure 3.

Author response: We have now added the table showing DEG list from RNA-seq in Figure 3 (Table S1).

Based on their RNA-seq data (which this reviewer could not verify due to lack of supporting

data) the authors have selected IAA19, SAUR40 and SAUR77 genes involved in auxin signalling for further analyses. The authors argue that these genes are HY5 targets by referring to a published data. What would be needed is to analyse HY5 binding to these and other key genes in the root and shoot at the experimental conditions.

Author response: This is another great suggestion. We have now added the ChIP data using 35S:HY5-GFP/*hy5-215* whole seedlings (Fig. S7D). Consistent with the published data (Burko et al., 2020; Fig S7C), HY5 enriches higher in the promoter region of IAA19, SAUR40, and SAUR77 at high ambient temperature.

Have the authors performed experiments for RNA-seq and the qRT-PCR experiment shown in Fig S7? Why do they see different results for BR-related genes? The authors show that unlike their RNA-seq data, expression of BR metabolic genes is temperature responsive. This temperature responsiveness was not observed in *hy5* mutant suggesting that HY5 may be essential. Why the authors did not include 35S:HY5 in this analysis to check if it is sufficient?

Author response: Martins et al. (2017) have shown that the root thermomorphogenesis is partly regulated by the BR-signaling pathway. However, the BR pathway genes did not show any differential regulation in our RNAseq experiment most likely due to low coverage in our experiment. Therefore, we performed qPCR experiments to test whether the BR pathway genes are defective in roots under our conditions (Old Fig. S7). We have now added the data showing *CPD*, *BAS1*, and *SOB7* transcript levels of 35S:HY5-GFP/*hy5-215* in both 22 and 28 using qPCR (Fig. S8A). Consistent with the qPCR result of WT and *hy5-215*, 35S:HY5-GFP/*hy5-215* showed altered gene expression of *CPD*, *BAS1*, and *SOB7* at high ambient temperature, indicating that HY5 is essential for temperature-mediated regulation of the BR metabolic genes.

The authors perform ChIP analysis on *BAS1* and *SOB7*, not *CPD*, for studying HY5 binding. The rationale, it seems, for not including *CPD* in this analysis was that they could not detect G-box element or its variants in the 2kb upstream sequence. Why limit at 2 kb? There have been examples of functional CREs downstream the TSS, within the gene body and even 3'.

In which tissue was the HY5 ChIP experiment performed? It would have been meaningful if these analyses were done under the experimental conditions with different temperatures.

Author response: In this study, we used the whole seedlings of 35S:HY5-GFP/*hy5-215* to perform the ChIP assays at different temperatures (Fig. S8). We did not perform ChIP assay for *CPD* as these data will not change the conclusions drawn in the manuscript. We performed two other BR-related genes just to show as an example that BR pathway is also directly regulated

by HY5 in root. Whether *CPD* is a direct target of HY5 can be tested in future.

Why was the HY5 western blot shown in Fig 4A and Fig S9 also not performed for root and shoot separately instead of only whole seedlings? The authors analyse expression of the HY5 transgene in shoot and root separately (Fig S8). This, along with other such discrepancies, makes this reviewer skeptical of the robustness of the authors analyses and wonder if they pick and present what is more convenient rather than providing a rigorous analysis!

Author response: We have now added the data showing separate Western blots using shoot and root samples (Figs. 6C, S10, S15). These results support the conclusions drawn in the manuscript.

The argument of “Phosphorylation of HY5 is necessary.....” is not sufficiently supported. At best they have shown that HY5-S35 has a modest effect on temperature response as shown in Fig 4D. In addition, the phosphomimic HY5-S35D did not show any phenotype. Therefore, the question of whether phosphorylation of HY5 is affected by temperature per se is not directly addressed. Moreover Fig. 4C, D and E should have used the *hy5* mutant as a control.

Author response: Fig. 4 shows that the HY-S36A failed to rescue the root elongation phenotype under high ambient temperature. In fact, the phenotype of HY5-36A is similar to that of *hy5-215*, suggesting that the phospho-null form is completely non-functional. On the other hand, the phospho-mimic form HY5-S36D complements just like wt HY5. We believe these data convincingly shows that phosphorylation of HY5 at S36 is critical for root thermomorphogenesis. We have now included *hy5-215* in Fig. 4C. Fig. 4D and E are GFP signals at two different temperatures, thus, *hy5-215* was not included.

The conclusions derived from *spaQ* mutant data (Fig 5) is not convincing as it seems *spaQ* is severely growth-defective, particularly very sensitive to 28°C. This suggests the important role SPAs play in general growth and development. Therefore, what is interpreted as a reduced or lack of root temperature responsiveness could well be a reflection of the defective development/growth in general rather than specific response of roots.

Author response: We agree with the reviewer that *spaQ* might have strong growth defects. However, there are many papers on the *spaQ* growth related phenotypes from various labs including our lab (Paik et al., 2019; Lee et al., 2020; Wang et al., 2021). In addition, what brings the specificity in our experiments is that addition of a single wt SPA1 can rescue the root phenotypes of *spaQ*, while introduction of a single amino acid mutant SPA1 failed to rescue the phenotypes (Fig. 5). These data suggest that the *spaQ* phenotypes may not be just severe

growth defects, but a specific response to high ambient temperature. Moreover, we have now added new data showing that the kinase activity of SPA1 on HY5 is temperature regulated (Fig. S11). In addition, these data in combination with Fig. 4 and S11 suggest that phosphorylation of HY5 by SPAs are necessary for root thermomorphogenesis.

The immunoblot data shown in Fig 6 is unclear due to what appears to be non-specific hybridizations and the lack of proper controls to unambiguously provide confidence in the data. There is no size marker or a negative control reagent (such as pif4 or hy5 mutants). Nor is there any loading control. These issues need to be clarified.

Author response: We have now added the data showing PIF4 and HY5 blot with proper negative controls and size marker (Fig. 6C). The new data is consistent with previous conclusion that HY5 is enriched in the root sample while PIF4 is enriched in the shoot samples under high ambient temperature.

Finally, the model proposed by the authors and therefore the schematic representation in Fig 6 is not fully supported. A clear explanation of how antagonistic transcription factors PIF4 and HY5 spatially uncouple their activities as proposed here to control temperature responsiveness in shoot and root respectively is not amply supported by experimental evidence.

In summary, the data presented here and the discussion thereafter, although interesting, are preliminary in its current form. The role of HY5 in root elongation and particularly temperature response is known. A rigorous follow up of what appears to be preliminary and less than rigorous analysis, to this reviewer, here could make this an interesting work that will unravel the molecular mechanism by which temperature controls root growth. It is important, however, to carefully address the conceptual and technical flaws.

Author response: We believe that the additional data added in this revision will help address the concerns raised by this reviewer.

Reviewer #2 (Remarks to the Author):

Authors for this manuscript demonstrate that two distinct transcription factors regulate thermo responses spatially. PIF4 is the key player regulating shoot responses to high ambient temperature. HY5 is the one regulating root thermo responses. In addition, both HY5 and PIFs are involved in regulating hypocotyl thickness in response to high ambient temperature. Their tissue-specific accumulation requires SPA1 or all four SPAs. They also show that root and shoot respond to high ambient temperature differently with only a small set of overlapping.

Major

1. Figures 4A and S9, separate root and shoot blots are needed to reflect possibly a large difference. In addition, normalization is preferred to a protein band through immunoblotting. The authors should also acknowledge that GFP fusion may not fully reflect the accumulation of a fusion protein since GFP tends to make a fusion protein much stable.

Author response: We have now added the Western blot data of shoot/root samples using the 35S:HY5-GFP/hy5-215 in Fig. S10. These data show that HY5-S36A version is more unstable than wt HY5 and HY5-S36D versions in both shoot and root samples. However, the root samples display much lower level of HY5-S36A than the shoot samples. We have also added a sentence raising the issue that the GFP fusion may not truly reflect the native HY5 levels.

2. In Figure 6C, better quality immunoblot of PIF4 is required and new HY5 immunoblot if possible. To show the specificity of the antibody, protein preparations from a pif4 mutant and a hy5 mutant should be loaded side by side.

Author response: We have now added the data showing PIF4 and HY5 blot with proper negative controls and size marker (Fig. 6C).

3. Are there two bands for PIF4 in shoot tissue and two bands of HY5 in root tissue? The fast migration band may be the phosphorylated band.

Author response: It has been shown that phosphorylation of PIF4 by SPA1 promotes accumulation of PIF4 at high ambient temperature (Lee et al., 2020). The new data show that PIF4 still has multiple bands indicative of phosphorylation, but HY5 did not show a band shift in the root sample. In our experimental conditions, native HY5 do not show a band shift; thus, this negative data preclude any conclusion about the phosphorylation status of HY5 in the root samples.

4. For Figure 6D, authors are encouraged to add the function of HY5 in thermo-induced hypocotyl thickness to the model. How is HY5 involved in thermo-induced hypocotyl thickness while its transcription is mostly thermo-induced in the root tissue? Its protein accumulation follows a similar pattern. Is the background level of HY5 sufficient to trigger the response?

Author response: HY5 is still expressed in the shoot samples and thus, may contribute to regulate hypocotyl thickness under high ambient temperature. As the reviewer suggested, the background level might be sufficient to regulate this response.

5. Both PIF4 and HY5 are involved in thermo-induced hypocotyl thickness. This is a very specific phenotype. Authors could discuss how they likely work together.

Author response: This has been discussed in the discussion section.

Minor

1. On page 8, change three different HY5 to three different HY5 forms.

Author response: It has been corrected.

2. In Figure 5 legend, the root length ration should be corrected to the root length ratio.

Author response: It has been corrected.

Reviewer #3 (Remarks to the Author):

This manuscript by Lee et al. reports the interesting spatial regulation of Arabidopsis thermomorphogenesis by HY5 and PIF4. PIF4 has been shown to act as a hub transcription factor to regulate the shoot responses to high ambient temperature, while in this study, the authors showed that HY5 acts as a key transcription factor to regulate root responses to high ambient temperature. The authors further showed that HY5 phosphorylation is required to stabilize HY5 in the root to promote primary root elongation at high ambient temperature, and that the kinase activity of SPA1 is necessary for root thermomorphogenesis. Thus, PIF4-SPA module and HY5-SPA module play pivotal roles in shoot and root, respectively, in plant response to high ambient temperature. Together, the data in this manuscript suggest that plants employ distinct combinations of the SPA-PIF4-HY5 module to regulate tissue-specific thermomorphogenesis.

In general, the data presented in this manuscript are novel and interesting, and contribute to a better understanding of high ambient temperature-induced architecture changes of sessile plants. Here are my suggestions to improve this manuscript.

1. Fig 2 shows that phyB regulates both above and below-ground tissues at high ambient temperature. Since phyB has been shown to be a thermosensor, it will be helpful to reflect the role of phyB in the model (shown in Figure 6D).

Author response: phyB is added in the model now.

2. Since the kinase activity of SPAs are essential for shoot and root thermomorphogenesis, how the protein levels or kinase activities of SPAs are regulated by high ambient temperatures? Also, it will be interesting to investigate whether HY5 phosphorylation is regulated by high temperature.

Author response: This is an excellent suggestion. As discussed above, we have now performed the kinase assay at both normal and high ambient temperature. The results show that the kinase

activity of SPA1 on HY5 is stimulated by high ambient temperature. We have also previously shown that the protein level of SPA1 is not regulated by high ambient temperature, but the mRNA of *SPA1* is down-regulated at 28C compared to 22C (Lee et al., 2020). However, this is only for *SPA1* and the transcription of *SPA2/3/4* is not regulated by high ambient temperature. We have also recently shown that all four SPAs can directly phosphorylate HY5 *in vitro* and they are necessary for the phosphorylation of HY5 at S36 *in vivo* (Wang et al., 2021). The phosphorylated form is more stable than the unphosphorylated form of HY5 (Figs. 4D, E; S10). Overall, these data suggest that SPAs might phosphorylate HY5 more at high ambient temperature to stabilize HY5 to promote root thermomorphogenesis.

3. Fig 6A and C show that the elevated HY5 protein level in root at high ambient temperature may be caused by upregulation of its gene expression. So how the authors define the contribution of SPA1-mediated phosphorylation in regulating HY5 protein abundance at high ambient temperature?

Author response: This is an important point. SPA1 directly phosphorylates PIF4 to stabilize PIF4 at high ambient temperature post-translationally (Lee et al., 2020). Since SPA1 kinase activity is stimulated by high temperature (Fig. S11), SPA1-mediated phosphorylation of HY5 might also contribute to the stability of HY5 in the root sample. However, we do not have sufficient data to dissect the individual contribution at this time.

Minor points:

1. Fig 3A: 2209 of HY5 regulated genes in total does not equal 665+418.

Author response: This has been corrected.

2. Statistical analyses are lacking in Fig 4B and 4G.

Author response: We have now added the data showing one-way ANOVA analyses with Tukey's HSD test for Fig 4E (Originally 4G). Due to other reviewer's suggestions for tissue specific GFP blot, we replaced 4B to S10.

3. P4: "Martins et al., 2017" should be replaced by the number "29".

Author response: This has been corrected.

4. P5, first paragraph: Fig 4A, B and C should be appropriately cited.

Author response: This has been corrected.

5. P6, third paragraph: "Fig. S6B" should be Fig. S6A.

Author response: This has been corrected.

6. Protein markers are suggested to be shown for Fig 6C.

Author response: We have now added the data showing PIF4 and HY5 blot with proper negative controls and size marker (Fig. 6C). In this blot, we have not added a protein marker as there are internal loading controls (indicated by *, ** and ***) in these blots.

REVIEWER COMMENTS

Reviewer #1 (Remarks to the Author):

The revised manuscript by Lee et al. has aimed to address most of the concerns raised by this reviewer in the first review.

New data has been added and most, if not all, of the concerns have been addressed to a satisfactory level. Some of the interpretations have been appropriately modified. This has made the manuscript and its arguments more convincing.

I have reservations on the immunoblot data shown in Fig S15, which was provided in response to a comment in the initial review. If properly normalised, there might not be any clear difference between HY5 protein levels in the root in LD and SD conditions. SD samples have significantly reduced (perhaps only half the amount) of loading as opposed to LD - clearly visible if looked at the non-specific band.

Reviewer #2 (Remarks to the Author):

The authors have properly addressed my comments.

Reviewer #3 (Remarks to the Author):

The authors have satisfactorily addressed my concerns.

Response to Reviewer's Comments

Reviewer #1 (Remarks to the Author):

The revised manuscript by Lee et al. has aimed to address most of the concerns raised by this reviewer in the first review.

New data has been added and most, if not all, of the concerns have been addressed to a satisfactory level. Some of the interpretations have been appropriately modified. This has made the manuscript and its arguments more convincing.

I have reservations on the immunoblot data shown in Fig S15, which was provided in response a comment in the initial review. If properly normalised, there might not be any clear difference between HY5 protein levels in the root in LD and SD conditions. SD samples have significantly reduced (perhaps only half the amount) of loading as opposed to LD - clearly visible if looked at the non-specific band.

Author response: We have repeated the blot and also used tubulin as a loading control. The new data show that HY5 is specifically upregulated in root tissue under LD conditions. Hope these data satisfy the reviewer's concern.

Reviewer #2 (Remarks to the Author):

The authors have properly addressed my comments.

Author response: Thank you very much.

Reviewer #3 (Remarks to the Author):

The authors have satisfactorily addressed my concerns.

Author response: Thank you very much.